# Effect of Clindamycin on Intestinal Microbiome and Miltefosine Pharmacology in Hamsters Infected with *Leishmania infantum*

**DOI:** 10.3390/antibiotics12020362

**Published:** 2023-02-09

**Authors:** Ana Isabel Olías-Molero, Pedro Botías, Montserrat Cuquerella, Jesús García-Cantalejo, Emilia Barcia, Susana Torrado, Juan José Torrado, José María Alunda

**Affiliations:** 1ICPVet, Department of Animal Health, School of Veterinary Sciences, Complutense University of Madrid, 28040 Madrid, Spain; 2Genomics Unit, Research Assistance Center of Biological Techniques, Complutense University of Madrid, 28040 Madrid, Spain; 3Department of Pharmaceutics and Food Technology, School of Pharmacy, Complutense University of Madrid, 28040 Madrid, Spain; 4Institute of Industrial Pharmacy UCM, School of Pharmacy, Complutense University of Madrid, 28040 Madrid, Spain

**Keywords:** miltefosine, clindamycin, visceral leishmaniasis, intestinal microbiome, *Leishmania infantum*

## Abstract

Visceral leishmaniasis (VL), a vector-borne parasitic disease caused by *Leishmania donovani* and *L. infantum* (Kinetoplastida), affects humans and dogs, being fatal unless treated. Miltefosine (MIL) is the only oral medication for VL and is considered a first choice drug when resistance to antimonials is present. Comorbidity and comedication are common in many affected patients but the relationship between microbiome composition, drugs administered and their pharmacology is still unknown. To explore the effect of clindamycin on the intestinal microbiome and the availability and distribution of MIL in target organs, Syrian hamsters (120–140 g) were inoculated with *L. infantum* (10^8^ promastigotes/animal). Infection was maintained for 16 weeks, and the animals were treated with MIL (7 days, 5 mg/kg/day), clindamycin (1 mg/kg, single dose) + MIL (7 days, 5 mg/kg/day) or kept untreated. Infection was monitored by ELISA and fecal samples (16 wpi, 18 wpi, end point) were analyzed to determine the 16S metagenomic composition (OTUs) of the microbiome. MIL levels were determined by LC-MS/MS in plasma (24 h after the last treatment; end point) and target organs (spleen, liver) (end point). MIL did not significantly affect the composition of intestinal microbiome, but clindamycin provoked a transient albeit significant modification of the relative abundance of 45% of the genera, including *Ruminococcaceae UCG-014*, *Ruminococcus 2*; *Bacteroides* and *(Eubacterium) ruminantium group*, besides its effect on less abundant phyla and families. Intestinal dysbiosis in the antibiotic-treated animals was associated with significantly lower levels of MIL in plasma, though not in target organs at the end of the experiment. No clear relationship between microbiome composition (OTUs) and pharmacological parameters was found.

## 1. Introduction

Leishmaniases are parasitic diseases transmitted by sandflies (Diptera, Psychodidae) and caused by protozoan species from the genus *Leishmania* [1]. Visceral leishmaniasis (VL), caused by *L. donovani* and *L. infantum*, is the most severe disease [2,3] and only second to malaria as the most lethal neglected tropical disease (NTD). An estimated 50,000 to 90,000 new cases of VL occur worldwide annually [4]. Leishmaniasis, once considered a tropical disease, is widely distributed (over 80 countries) and its geographical distribution is increasing [5,6] and is considered a global challenge both in the medical and veterinary arena [7,8]. Anthropogenic climatic change facilitating the wider distribution of sandfly vectors and the disease [9,10,11,12] is considered an important factor in the expansion of leishmaniasis. Moreover, emerging transmission patterns such as solid organ transplants [13,14,15,16,17,18,19,20,21] and coinfections in immune-suppressed patients [22,23,24] have also been incriminated and there is a need for continuous effort to reduce the impact of the disease [25].

Environmental control of leishmaniasis is, in most cases, unfeasible, particularly in the case of non-zoonotic transmission. Since no vaccines against human VL are available and those marketed against CanL have shortcomings [26], chemotherapy of affected individuals is by far the most used control system. The therapeutic arsenal to treat VL is scarce and has important drawbacks, including toxicity, lack of efficacy, emergence of resistances and low compliance due to the administration route and required hospitalization for medication with some drugs [27,28,29,30,31]. No new chemical entities have been identified and current chemotherapy is based on the use of drug combinations, low-toxicity presentations of the currently available drugs (e.g., nanoformulations) or drug repurposing, among other strategies [32,33,34,35]. Thus, there is an urgent need for drugs against leishmaniasis [36].

Miltefosine (hexadecyl phosphocholine) (MIL) is the most recent antileishmanial drug marketed. The drug is still the only oral medication for leishmaniases, with efficacy comparable to that of antimonials. It was originally developed as an anticancer medication [37] and besides its antiprotozoal activity, both antibacterial and antifungal activities of MIL have been reported [38,39,40]. It is generally well tolerated although gastrointestinal disturbances have been described and it is considered by the WHO as a first line antileishmanial drug in those areas where resistance to pentavalent antimonial is present [41,42,43,44,45] (e.g., northern India, Nepal). The pharmacology of MIL is not completely known [46,47,48] although its efficacy has been found to be strongly correlated to the levels of MIL reached in plasma and target organs [49]. Reported clinical failures of MIL in the treatment of VL and its poor efficacy in pediatric medicine could be due to low drug exposure [49,50] because of inadequate dosage or reduced intestinal absorption.

The relationship between the microbiome and healthy/diseased conditions of humans, domestic animals or surrogate models is a well-represented research area and the interaction is considered one of the most critical factors that determine the outcome of infection [51]. Several studies have been carried out in surrogate models and in VL patients by cross-sectional stool sampling [52,53,54,55]. However, the interaction between microbiome and drugs administered has been less well studied [56,57]. Moreover, comorbidity—patients being affected by more than one disease—is common in leishmaniasis [58,59,60,61] and the impact of antibiotics on the intestinal microbiome has been recognized [62,63]. In the case of VL and MIL, this interaction has not been explored, as far as we know, despite being the only antileishmanial drug of oral administration.

On these grounds, we have examined, in hamsters experimentally infected with *L. infantum*, the interaction between an antibiotic treatment (clindamycin), MIL pharmacology (absorption and biodistribution) and composition of the intestinal microbiome.

## 2. Results

### 2.1. Orally Administered Miltefosine (MIL) Does Not Significantly Modify the Main Composition of the Intestinal Microbiome of Syrian Hamsters Infected with Leishmania infantum

All inoculated animals developed an *L. infantum*-specific serum IgG response, without differences among infected hamsters, whereas uninfected control animals were negative for the entire experiment (Appendix A: Serum-specific anti-*Leishmania infantum* IgG response of *L. infantum*-inoculated and control hamsters throughout the experiment determined by ELISA). Investigation of the composition of intestinal microbiome carried out in week 16 of the experiment, before administering any therapeutic agent, allowed us to identify 21 phyla, 154 families (including 16 assigned to different uncultured families) and 344 genera (including 57 assigned to different uncultured genera) in the experimental hamsters. The Shannon index in uninfected hamsters was 3.26 for G2, and 3.36 and 3.21 for L. infantum-infected animals (G4 and G5). The most abundant phyla in the animals were *Firmicutes* and *Bacteroidetes*, representing ca. 95% of all OTUs identified, followed by *Proteobacteria*, *Verrucomicrobia, Patescibacteria, Cyanobacteria* and *Epsilonbacteraeota*, with values lower than 1.5% each (Figure 1 and Appendix A: Phylum abundance (%) in groups G2, G4 and G5 at week 16). Standard statistical analysis did not show significant differences in phylum abundance among groups (*p* > 0.05). Differential abundance analysis (DAA) considering FDR *p <* 0.05 of OTUs confirmed the absence of differences in phyla in week 16, among the hamster groups (G2, G4, G5).

This over-dispersed pattern of abundance was also found in families since three families (*Ruminococcaceae, Muribaculaceae* and *Lachnospiraceae*) represented ca. 80% of the OTUs and the 25 most abundant families accounted for ≥99% of all identified families (Appendix A: Abundance (%) of the 25 most represented families in groups G2, G4 and G5 at week 16). There were no significant differences, considering these 25 families, among the experimental animals irrespective of the infection status (week 16: G2 vs. G4, G5). From the genera identified, *uncultured bacterium-07*, *Lachnospiraceae NK4A136 group*, *uncultured-15*, *Ruminococcaceae UCG-014*, *uncultured-13*, *Ruminoclostridium 6*, *Alloprevotella*, *Ruminococcus 1* and *(Eubacterium) ruminantium group* were the most abundant (ca. 60% of all OTUs identified). All other genera had scarce representation since the 40 most abundant reached ca. 90% of all taxa detected (Appendix A: Abundance (%) of the 25 most represented genera in groups G2, G4 and G5 at week 16). Statistical analyses performed on these 40 genera did not show any differences related to the infection status of the animals (G2, G4, G5 16w, *p* = 0.965). In addition, we could not find any significant difference in phylum abundance in the uninfected group (G2) before (16 wpi) or after receiving MIL treatment (18 wpi) (see Figure 2). In the uninfected group, MIL treatment elicited a significant increase in *Bacteroidaceae* (*p* = 0.00040) although the average abundance of this family ranged, before treatment, from 0.30–0.49%. Uninfected hamsters increased their values of *Ruminococcus 2* after medication with MIL (G2 16w vs. G2 18w, FDR *p* = 0.00036) although the relative abundance of this genus was very low. Similarly, MIL treatment did not elicit notable modifications in the microbiome composition of infected hamsters (G4) since only minor differences were found in the OTUs determined (Appendix A: Phylum abundance (%) in groups G2 and G4 at weeks 16 and 18, Appendix A: Differential abundance (%) analysis of the 25 most abundant families detected in groups G2 and G4 at weeks 16 and 18 and Appendix A: Differential abundance (%) analysis of the 40 most abundant genera detected in groups G2 and G4 at weeks 16 and 18). Shannon index values of the microbiome from infected hamsters did not show relevant variations after treatment with MIL alone (G4 18w: 3.29).

### 2.2. Clindamycin Elicits a Deep Although Transient Modification of the Intestinal Microbiome of Hamsters Infected with Leishmania infantum and Treated with Miltefosine (MIL)

Treatment with the antibiotic did not affect the relative abundance of the most common intestinal phyla (*Bacteroidetes* and *Firmicutes*) although the medication elicited some modifications of less abundant phyla. Thus, *Deferribacteres* showed a 7.5× reduction (*p* = 0.0072) whereas *Verrucomicrobia* and *Fusobacteria* had higher abundance (7× and 60×, respectively (*p* = 0.0432; *p* = 3.929 × 10^−7^)) in the animals treated with clindamycin + MIL (G5) compared to the hamsters only receiving MIL (G4) (18 wpi) (Appendix A: Differential abundance (%) analysis of the phyla detected in groups G4 and G5 at week 18 and PF). The alteration of the microbiome elicited by the antibiotic was transient and at the last sampling (FP) no significant differences in the relative abundance of phyla between the two experimental groups were found. Accordingly, ubiquitous bacterial families in mammals were the most abundant OTUs in both experimental groups after treatment (week 18 pi): *Muribaculaceae* (*Bacteriodetes*) (26.77%: G4; 23.92%: G5) and two *Firmicutes* (*Ruminococcaceae*: 29.13%, G4 vs. 19.32%, G5; *Lachnospiraceae*: 27.73%, G4 vs. 23.99%, G5). However, treatment with the antibiotic significantly modified the relative abundance of several less represented families (<1% OTUs) by increasing (*Bacteroidaceae*: 24×, *p* = 1.8104 × 10^−11^; *uncultured-04*: 14×, *p* = 1.0598 × 10^−5^; *Tannerellaceae*:10×, *p* = 1.0598 × 10^−5^; *Enterobacteriaceae*: 37×, *p* = 0.00126512; *Akkermansiaceae*: 8×, *p* = 0.03541358; *Fusobacteriaceae*: 55×, *p* = 2.8071 × 10^−6^; *Defluviitaleaceae*: 10×, *p* = 0.00262074; *Burkholderiaceae*: 12×, *p* = 4.1359 × 10^−8^) or, to a lesser extent, decreasing (e.g., *Clostridiales vadinBB60 group*: ×16, *p* = 0.02534581; *Deferribacteraceae*: 6×, *p* = 0.0315025) (Appendix A. Differential abundance (%) analysis of the families with an abundance >0.01% detected in groups G4 and G5 at week 18 and PF).

Analysis of genera showed that, in week 18, treatment with clindamycin caused a generalized and significant modification of the intestinal microbiota of infected and treated hamsters. Thus, considering those genera with ≥0.5% OTUs, nearly half of them (45%) modified their abundance and 27% showed FDR *p* < 0.05. The abundance of 24 genera with abundance >0.1% was significantly different when G4 and G5 groups were compared in week 18 (Appendix A: Differential abundance (%) analysis of the genera with significant abundance changes (FDR *p* > 0.05) detected in groups G4 and G5 at week 18 and PF; Figure 2). It is noteworthy to indicate that the genus pattern modified in antibiotic-treated hamsters affected some abundant taxa (e.g., *Ruminococcaceae UCG-014*: 5.47157%, G4 vs. 0.80430%, G5; *Ruminococcus 2*: 5.32% G5 vs. 0.14% G4; *Bacteroides*: 10.71% G5 vs. 0.56% G4: *(Eubacterium) ruminantium group*: 4.16895%, G4 vs. 0.24170% G5); these changes were reflected as a modified abundance bacteriogram (Figure 2). Despite the modification of the microbiome, no reduction of Shannon index value was observed in the hamsters treated with the antibiotic (G5 16w: 3.21 vs. G5 18w: 3.35).

### 2.3. Pharmacokinetics (PK) and Biodistribution of Miltefosine (MIL)

Plasma levels of MIL in the hamster groups are shown at Figure 3. Daily administration of the drug allowed plasma levels ranging from 19.63 ± 2.51 μM (G5) to 25.41 ± 5.62 μM (G4) to be reached, 24 h after the last treatment. Significant differences between groups (*p* = 0.0291) were observed due to the higher levels found in G4 hamsters compared to those infected and treated with MIL and clindamycin (G5) (*p* = 0.0187). Determination at the end point confirmed the differences between groups (*p* = 0.0069) with lower values in clindamycin-treated hamsters compared to the animals only treated with the alkyl phospholipid (G4 vs. G5, *p* = 0.0288). For their part, the two groups not subjected to the medication with the antibiotic did not show significant differences between them (*p* = 0.1518).

Estimated half-life values of MIL showed differences between experimental groups (*p* = 0.0311). The mean value in uninfected and treated animals was 59.5 ± 8.31 h, slightly higher than that found for those infected and treated with MIL (49.67 ± 6.32 h; *p* = 0.0308) or the infected animals treated, in addition, with clindamycin (51.18 ± 5.49 h; *p* = 0.0495); with no differences among infected hamsters (*p* > 0.05). At the end of the experiment, the mean levels of MIL in the liver were not different between G2 vs. G4 (*p* = 0.1020) and G2 vs. G5 (*p*= 0.1385) or the infected vs. uninfected animals (*p* = 0.6521) (Table 1). Comparable results were found when considering the MIL concentration in the spleen.

No differences were found in liver/plasma and spleen/plasma ratio MIL values. Liver/spleen ratio of MIL was higher in the antibiotic-treated group although with high individual variation (L/SMIL = 4.95 ± 3.9) and therefore the differences between groups were not significant.

## 3. Discussion

In our experiment, sequential sampling confirmed the absence of significant modifications of the intestinal microbiome of hamsters, irrespective of the *L. infantum* infection [55], as assessed by the comparable OTUs found in all animals in week 16. These findings support the limitations of cross-sectional studies in VL patients [52] and the need for using sequential samplings. The mechanistic basis of the antileishmanial activity of MIL is not completely elucidated [64,65,66] although it has been shown that MIL inhibits phosphatidylcholine (PC) biosynthesis [67,68,69]. PC is the major membrane-forming phospholipid in eukaryotes, being estimated that it is present in about 15% of the domain Bacteria [70]. Therefore, some effect on the intestinal microbiome would be expected after MIL medication and gastrointestinal disturbances have been described in humans [41] and dogs [71,72]. However, our results showed that predominant OTUs were not affected after treatment with MIL since in both G2 (uninfected control group) and G4 animals (*L. infantum*-infected hamsters), *uncultured bacterium*, *Lachnospiraceae NK45A136 group*, *Ruminococcaceae UCG-014* and *uncultured* were the prevalent genera and no differences were found in higher taxa. The significance of the elevation of *Ruminococcus 2* (*Clostridiales*) in MIL-medicated hamsters is not known since there is still no understanding of the role of *Ruminococcus* spp. in their respective hosts [73]. Whether or not the scarce impact on the microbiome observed in our experiment is due to the short period of treatment (7 days) or the dose administered (5 mg/kg/day) should be further investigated since standard treatment in target species (humans, dogs) is longer and with lower doses (28 days, 2–2.5 mg/kg/day).

The lincosamide antibiotic clindamycin possesses activity against most Gram-positive bacteria but has virtually no activity against aerobic Gram-negative bacteria [74,75] and induces a marked modification of the intestinal microbiome of the individuals treated [76,77,78,79,80,81,82,83,84]. The profound dysbiosis elicited by clindamycin was confirmed despite the much lower antibiotic dosage employed (PO 1 mg/kg, single dose) compared to other experiments in hamsters (e.g., PO 200 µg/animal, 50 mg/kg) [78,79] and mice (e.g., SC, 1.5 mg/day, 3 days) [76]. The described reduction of *Bacteriodetes* in hamsters [79] and mice [76] treated with the antibiotic was not observed in our experiment, whereas the reported rise of *Proteobacteria* in treated animals [76] was discreet. *Fusobacteria*, *Verrucomicrobia* and *Euriarchaeota* were the phyla displaying the highest relative increase in clindamycin + MIL-treated hamsters, but its actual significance is not known. Intestinal dysbiosis was more evident considering genera. Although we did not find any significant reduction of total *Firmicutes*, some of the major genera, *Ruminoclostridium, Ruminocccaceae UCG-014* and the (*Eubacterium*) *ruminantium* group, showed a significant reduction after clindamycin medication [76]. These authors did not report variations of *Akkermansia* but, as observed in mice [78], we also found higher abundance of this genus in clindamycin-treated hamsters. The relative increase in *Bacteroides*, *Parabacteroides* and *Ruminococcus 2* (Gram-positive) in the treated animals could be related to the low sensitivity and resistance of the genera from *Bacteroidetes* [85]. Treatment with this antibiotic has been related to a marked reduction of microbiome biodiversity [78,84] (Shannon index from 5 to <1: [78]; 1/3 of taxa lost: [84]). Results obtained in our case did not show significant differences in the index (~3) among the experimental groups one week after treatment (week 18) and at the end point of the experiment. This suggests a rapid recovery of the microbiome, in line with results in mice [77], and supports the resilience of the intestinal microbial community [84]. Comparison of the results should be cautiously considered given the different animal species studied, treatment dose and schedule, calendar of sampling besides the described variations depending on the diet, management and starting status of the microbiome [86,87,88].

Results obtained in the determination of some pharmacological parameters of MIL in hamsters, their relationship to the *L. infantum* infection and the intestinal dysbiosis elicited by the medication with clindamycin raise several questions. The pharmacology of MIL has been studied in humans and surrogate rodent models [64,89]. This drug is slowly absorbed upon oral administration [90] and <10% of the drug is eliminated through feces [91]. Thus, some type of interaction between the microbiome and MIL pharmacology would be expected. The duration of the experiment, besides ethical constraints, limited the number of plasma samples but our results point towards a modest effect of the deep clindamycin-induced intestinal dysbiosis on MIL absorption. We do not have an adequate explanation for the apparently better performance of infected hamsters (G4), compared to the uninfected and MIL-treated animals (G2) 24 h after the last treatment since no notable differences were found in OTU abundance between the animal groups. Significantly lower plasma levels of MIL in clindamycin-treated hamsters (G5) at the end point suggest that microbiome dysbiosis affects the availability of the phospholipid. The mechanistic basis of the lower levels of MIL reached in plasma of clindamycin-treated hamsters needs further research although partial degradation of MIL by phospholipase (D, C) bacterial activity [92,93] in the clindamycin-modified microbiome cannot be ruled out given the slow absorption of oral MIL [90]. In our experiment, although the concentrations of the drug in plasma and analyzed organs, especially the main target organs, were correlated, no significant differences were found among the animal groups. The lack of differences could possibly be due to the duration of the experiment, high bioavailability (>80% in rats and dogs) and long half-life of MIL [64]. All families of antibiotics, including aminoglycosides, have a profound effect on the composition of the intestinal microbiome [62], sometimes reducing the abundance of so-called beneficial microorganisms [81]. This reduction has also been observed with paromomycin therapy [94,95]. The combination of MIL and the aminoglycoside paromomycin [34] has been suggested in humans. Besides the potential harmful effect of the antibiotic on the intestinal microbiome [94,95], our results with lower levels of MIL in the plasma of clindamycin-treated hamsters suggest the need for studying the pharmacology of MIL in this combination therapy to minimize the risk of the appearance of MIL-resistant strains by subdosification. This is critical since to date MIL is the only available oral medication for leishmaniasis and the alternative frontline antileishmanial in endemic regions with visceral *Leishmania* resistant to antimonials [41,42,43,44,45].

## 4. Materials and Methods

### 4.1. Chemicals and Drugs

MIL, LC-MS grade methanol and water were purchased from Sigma-Aldrich (Madrid, Spain). Acetonitrile (Scharlab, Barcelona, Spain), ammonia 25% and triethylamine (Panreac, Madrid, Spain) and glacial acetic acid (Fisher Scientific, Madrid, Spain) were of analytical grade. MIL for oral administration was Milteforan (Virbac, Carros, France) (20 mg/mL, Lot 7MD7A) and clindamycin was from Vétoquinol (Paris, France) (75 mg/capsule, Lot E36246/A).

### 4.2. Parasites and Hamsters

Male Syrian hamsters (*Mesocricetus auratus*) (*n* = 22) were purchased from Janvier Labs (Marseille, France) (7–8 weeks old, 85–120 g) and subjected to quarantine. Briefly, animals had tap water ad libitum, and were fed with commercial pelleted food in polystyrene cages at animal facilities (Instituto de Investigación Hospital 12 de Octubre, Madrid, Spain). When hamsters reached 120–140 g live weight (lw), they were divided in a stratified way (live weight) and inoculated with *L. infantum* (MCAN/ES/ 96/BCN150) (*n* = 16) or kept as uninfected control animals (*n* = 6). An infective dose (106 promastigotes/hamster) was administered by IV retroorbital inoculation [96]. Infections were maintained for ca. 16 weeks (120–122 days postinfection, dpi) and infected hamsters were reallocated (balanced live weight) into 2 groups: Group 4 (*n* = 8): infected with *L. infantum* and treated with MIL (5 mg/kg/day, 7 days); and Group 5 (*n* = 8): infected with *L. infantum* and treated with a single dose of clindamycin (1 mg/kg) and MIL (5 mg/kg/day, 7 days). In addition, the uninfected control group (Group 2) received MIL 5 mg/kg/day, 7 days. All animals were euthanatized 7 days after the last treatment (132–136 dpi) (19 wpi). The number of animals was estimated to give a z-power of 0.8 and 95% level of significance. The number of uninfected control animals was reduced on ethical grounds and the previous experience of the group on this host–parasite model.

### 4.3. Follow-Up and Assessment of Infection

Animals were observed daily and weighed on day 0 (preinfection), 120–122 (initial day of treatment) and at the end point of the experiment (7 days after the last day of treatment). Blood samples were obtained from the cava vein under anesthesia with 2–4% isoflurane (0, 16 wpi) and by intracardiac puncture at the end point of the experiment (19 wpi). Serum and plasma samples, for immunological and pharmacological determinations, respectively, were used immediately or stored at −20 °C. Individual fecal samples from experimental hamsters were obtained on 16 wpi, 18 wpi and at the end point and stored at −80 °C until processing. Assessment of leishmanial infection status was determined by indirect ELISA; optical density (OD) cut-off (+/−) was established at mean preinfection values + 3 SD (13.74%). All determinations were performed at least in triplicate.

### 4.4. Determination of Miltefosine in Plasma and Target Organs’ Samples

Determination of MIL levels in plasma and organs (spleen, liver) was carried out following Dorlo et al. [97] with some modifications [65]. In brief, plasma samples (20 μL) were diluted in 1250 μL 0.9 M acetic acid, vortexed and centrifuged at 4000× *g*, 10 min. The supernatant was recovered and analyzed. After euthanasia, the organs were immediately extracted and weighed. Portions (ca. 0.02–0.04 g) were homogenized in 1 mL 0.9 M acetic acid (ca. 20,000 rpm for 30 s) (OMNI TH tissue homogenizer) followed by three freezing-and-thawing cycles in liquid N2 and a water bath at 37 °C. After assessing cell disruption, the mixture was vortexed and centrifuged at 3220× *g* for 20 min at 4 °C and the supernatant recovered and kept frozen until analysis. Extraction of MIL from the biological samples was performed by solid phase extraction (SPE) through Bond Elut PH (phenyl) SPE cartridges (Agilent). The cartridge was conditioned with 1 mL acetonitrile and then with 1 mL of 0.9 M aqueous acetic acid. Diluted plasma (1 mL) or 250 μL of the supernatants obtained after organ preparation was added, the cartridge was washed with 1 mL of methanol–water (1:1, *v*/*v*) and the analyte was eluted with 2 washes of 0.75 mL of 0.1% (*v*/*v*) triethylamine in methanol, filtered (0.45 μm PTFE filters) and injected into the analytical column. A density of 1 was assumed for spleen and liver when molar units were employed for these organs. Samples were analyzed by liquid chromatography (LC) coupled to a QQQ mass spectrometer equipped with a turbo ion spray source operating in positive ion mode (LCMS 8030, Shimadzu). Chromatographic separation was performed on a Gemini C18 analytical column (150 mm × 2.0 mm I.D., 5 μm particle size; Phenomenex) coupled with a C18 guard cartridge (4 × 2.0 mm; Phenomenex). Injection volume was 20 μL. Samples were delivered over 10 min at a flow rate of 0.2 mL/min through the analytical column at 45 °C. The mobile phase was composed of A (0.1% formic acid in water) and B (methanol). Mobile phase composition began with 0% B and was increased to 95% B in 3 min. The mobile phase was then maintained at 95% B for 2 min and decreased to 0% over the next 2 min, followed by re-equilibration with 0% B for 3 min before injecting the next sample. Quantification of MIL was performed by multiple reaction monitoring (MRM) mode to monitor the parent ion–product ion (*m*/*z*) of the analyte. Mass transitions of *m*/*z* 408.5 to 86.05 (CE = −37 V) were used for quantification and *m*/*z* 408.5 to 124.9 (CE = −30 V) for identification with a dwell-time of 100 ms. The calibration curve was determined by plotting the peak area of the analyte (Y) versus the nominal concentration (X) with least square linear regression. The limits of quantification (LOQ) and detection (LOD) were 1 ng/mL and 0.25 ng/mL, respectively, with a linear dynamic range between 1 and 500 ng/mL. Plasma samples spiked with 1, 10, 75 and 500 ng/mL MIL concentrations were run for QC. Between-run accuracy (%) for 1, 10, 75 and 500 mg/mL was −9.1, −3, 11.2 and −0.9, respectively. Between-run precision (%) for 1, 10, 75 and 500 mg/mL was 16.3, 9.6, 4.4 and 9.2, respectively. Within-run accuracy and within-run precision variations were lower than 10% for all QC. All analyses were carried out under ISO 9001:2008 quality management system certification.

### 4.5. Genomic Analysis of Microbiota

#### 4.5.1. Fecal Samples, DNA Extraction, 16S Metagenome Library Construction and NGS Sequencing

Fecal samples were obtained at three time points in the experiment (16 wpi, 18 wpi and at the end point) and preserved (−80 °C) until analyzed. In the first two samplings, individual hamsters were isolated and the feces were collected; at the end point of the experiment, the samples were taken from the rectum at necropsy. DNA extraction, NGS library construction and sequencing were performed at the Genomics Unit at the Complutense University of Madrid. In brief, total DNA from hamster fecal samples was extracted (DNeasy PowerLyser PowerSoil DNA Kit, Qiagen, Hilden, Germany) and DNA concentration was estimated with the Qubit 2.0 Fluorimeter. DNA libraries were prepared following the “16S Metagenomic Sequencing Library Preparation” manual from Illumina (Illumina, San Diego, CA, USA): the V3–V4 region of the prokaryotic 16S rRNA was amplified for each sample with primers containing the 341F and 805R sequences and Illumina-specific adapters. Two specific 8-nucleotide index and i5/i7 Illumina adapters were added to the previous amplicons in a second PCR amplification. DNA libraries were checked with the Bioanalyzer 2100 (Agilent Technologies, Palo Alto, CA, USA). A library was prepared by pooling equal amounts of the individual sample libraries. The library pool was sequenced in Illumina MiSeq equipment with 2 × 300 reads using the 600 cycle MiSeq Reagent Kit v3, as recommended by the manufacturer.

#### 4.5.2. Sequence Data Analysis

The FASTQ files containing the sequencing reads were analyzed using the CLC Genomics Workbench version 20.0.4 (QIAGEN Aarhus A/S, Aarhus, Denmark). Sequence data were trimmed using 0.05 as a limit for quality scores with 2 as the maximum number of ambiguities. The reads after trimming were analyzed using the CLC Microbial Genomics Module version 20.1.1. The optional merge paired reads method was run with default settings (mismatch cost = 1; minimum score = 40; gap cost = 4 and maximum unaligned end mismatch = 5). Sequence reads were clustered and chimeric sequences detected using an identity of 97% as the operational taxonomic unit (OTU) threshold. Reference OTU data used in the present study were downloaded from the SILVA database v132 for 16S rRNA [98]. The Shannon diversity index was calculated considering the assigned species. The raw sequencing data were deposited in the NCBI Sequence Read Archive [99] (BioProject ID: PRJNA843999) (https://www.ncbi.nlm.nih.gov/bioproject/PRJNA843999, (accessed on 25 November 2022).

### 4.6. Statistical Analysis

Experimental groups were included in a larger experiment and the number of animals was chosen to give a z-power of 0.8 and 95% level of significance. Numerical values are, unless otherwise stated, mean ± standard deviation or mean ± standard error. Statistical analysis from ELISA and pharmacology included parametric and non-parametric tests (1 w and 2 w ANOVA, Mann–Whitney test, Student’s t-test) and the level of significance was set at *p* ≤ 0.05. Taxonomic comparison between groups was performed with the Differential Abundance Analysis Tool from the CLC Microbial Genomics Module. The table of OTUs generated by the CLC Microbial Genomics Module from each microbiome classified at phylum, family or genus levels was used as the input. Unless otherwise stated, only changes with at least ±2-fold (+/−) in present taxa and false discovery rates (FDRs) with adjusted *p* value ≤ 0.05 were considered as significant. Figures were prepared with GraphPad Prism 6.0 and Microsoft Excel.

## 5. Conclusions

This research has addressed, for the first time, the relationship between the intestinal microbiome modification induced by a broad-spectrum antibiotic (clindamycin) in hamsters experimentally infected with *L. infantum* and several pharmacological parameters of MIL. Our results suggest that oral MIL treatment, with the dose and schedule used, apparently has scarce impact on the intestinal microbiome irrespective of the infection status of the animals. Modification of its composition by a single dose of clindamycin was profound albeit transient, which points towards the resilience of the intestinal microbiome. Both the infection status (*L. infantum*) of the animals and antibiotic medication reduced the plasma levels of MIL but not the concentration of the drug in target organs at the end point. Availability of the phospholipid, with standard treatment schedules, is critical for the treatment of leishmaniasis since there is high correlation between exposure of *Leishmania* to MIL and efficacy [49,50]. The authors are aware of the preliminary nature of the research performed, the complex interactions and the still fragmentary knowledge of the actual role played by bacterial and non-bacterial components of the intestinal microbiome. More experiments in surrogate models, under controlled conditions, with higher numbers of animals and longer duration of treatments, closer to the standard therapeutic regime in target species (humans, dogs), are needed to establish a causal framework. Despite these limitations, results obtained in the most advanced rodent model of VL provide a baseline for future and more refined studies unraveling the relationship between absorption of MIL (and other orally administered drugs) and antibiotic-induced microbiome dysbiosis. Drug combinations to treat leishmaniasis with antibiotics should be cautiously considered, given the effect of clindamycin on MIL plasmatic levels, and the need for studying pharmacological parameters of hexadecyl phosphocholine is strongly stressed to reduce the potential risk of emergence of resistant *Leishmania* strains against the only orally administered antileishmanial drug.

## Figures and Tables

**Figure 1 antibiotics-12-00362-f001:**
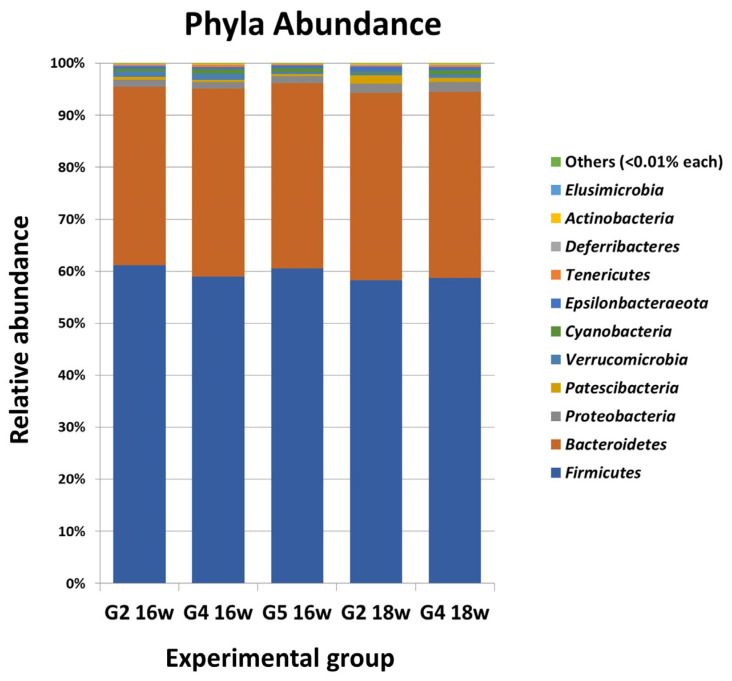
Mean relative abundance (%) of the most abundant phyla in the intestinal microbiome of Syrian hamsters infected for 16 weeks with *Leishmania infantum* (G4 16w, G5 16w), infected and treated with MIL (G4 18w) and uninfected control animals before or after MIL treatment (G2 16w, G2 18w).

**Figure 2 antibiotics-12-00362-f002:**
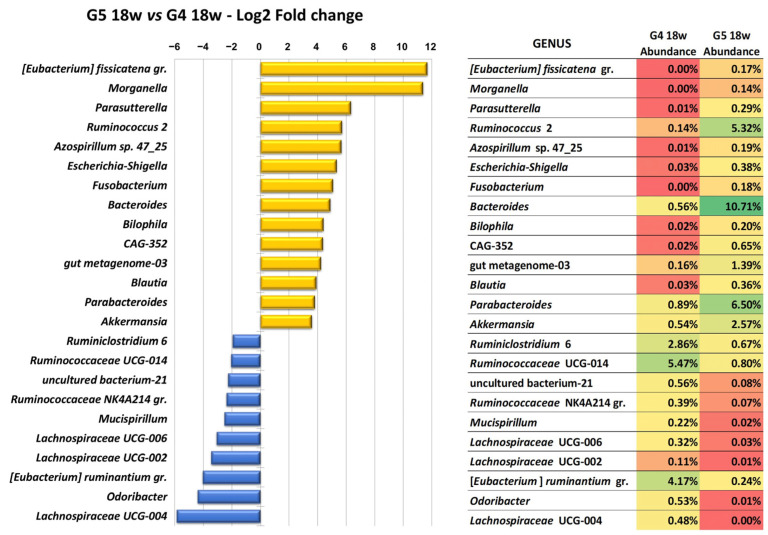
**Left:** Genera with significant change in abundance (FDR *p*-value < 0.05) in hamsters infected with *Leishmania infantum* treated with MIL (G4 18w) or MIL + clindamycin (G5 18w). Changes (log2 fold) in genera with abundance >0.1% in any condition are shown. **Right**: Values given correspond to the mean values of genera abundance from each experimental group. Color correspond to the approximate abundance of taxa, from more abundant (green), medium (yellow), to less abundant (red) genera.

**Figure 3 antibiotics-12-00362-f003:**
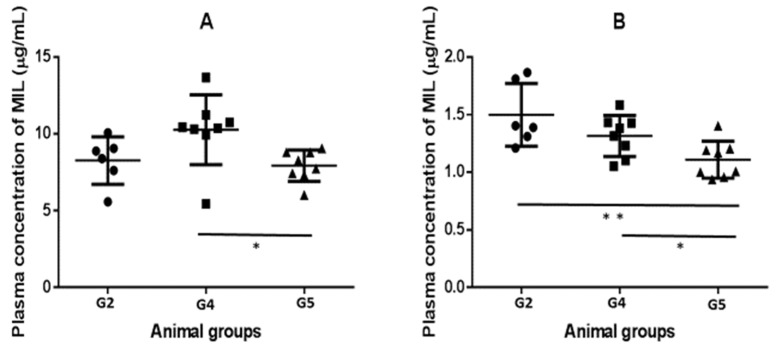
Levels of MIL in plasma from experimental hamsters, 24 h after the last day of treatment (**A**) and at the end point of the experiment (**B**). Solid circles: G2: non-infected and treated with MIL; Solid squares: G4: *Leishmania infantum* infected and treated with MIL; Solid triangles: G5: hamsters infected with *L. infantum* and treated with clindamycin and MIL. Individual data (dots) and mean and standard deviation from each group are given. Significant differences between groups: * *p* < 0.05; ** *p* < 0.01.

**Table 1 antibiotics-12-00362-t001:** Concentration of MIL (µM) in target organs of experimental animals and organ/plasma ratio at the end point of the experiment (mean ± SD).

Animal Group	Liver (µM)	Spleen (µM)	Liver/Plasma Ratio	Spleen/Plasma Ratio
G2	3.10 ± 0.70	0.92 ± 0.41	0.87 ± 0.30	0.27 ± 0.16
G4	2.43 ± 0.70	0.85 ± 0.37	0.76 ± 0.24	0.27 ± 0.14
G5	2.58 ± 0.54	0.71 ± 0.31	0.95 0.17	0.26 ± 0.12

## Data Availability

Data are included in the manuscript and Appendix A. The raw sequencing data from the microbiome were deposited in the NCBI Sequence Read Archive (BioProject ID: PRJNA843999) (https://www.ncbi.nlm.nih.gov/bioproject/PRJNA843999, (accessed on 25 November 2022).

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
