# Peer review of "Effect of Clindamycin on Intestinal Microbiome and Miltefosine Pharmacology in Hamsters Infected with Leishmania infantum"

_antibiotics, 2023, doi:10.3390/antibiotics12020362_

Round 1

Reviewer 1 Report

1.       L.infantum should be full name initially

2.       Abstract need overhauling; information's about the background and objectives are given but no detailed methods, quantitative results are provided. Generalized statements and vogue terms must be avoided

3.       In the introduction; the global burden of leishmaniasis, unavailability of effective therapeutics and search for more effective drugs or combinations or repurposing should be targeted in the initial paragraph. Current information's are not enough

4.       Add most recent references and informations

5.       Ethical committee approval number is required for animal studies

6.       Add relevant references to the methods used

7.       Revise discussion in the light of relevant work published previously

Reviewer 2 Report

The manuscript presented by Ana I. Olías-Molero et al described that effect of miltefosine treatment on the intestinal microbiome of Syrian hamsters. Moreover, the authors also discussed the effect of clindamycin on the intestinal microbiome of hamsters treated with mitefosine. Most experiments are well controlled. Some conclusions are not support by evidence. Below are some comments to help strengthen the manuscript.

1.    Methods should definitively be improved to help understand the statistical analysis of this paper. Please add as many details as you can to allow a proper understanding of your concept.

2.    Legends are way too vague, do not emphasize the quality of their work don’t emphasize the quality of their work and make difficult to follow the flow and rational of the work.

3.    For Figure 3, Whether there were significant differences between the groups?

4.    English should be polished to improve quality of the work.

Reviewer 3 Report

The manuscript is about the Effect of clindamycin on intestinal microbiome and miltefosine pharmacology in hamsters infected with Leishmania infantum. The study was addressed in an integrated and rigorous manner. The outcomes are of utmost importance in the context of Leishmaniasis treatment. Generally, the writing and arguments in the paper are technically valid and the text is easy to read. This work constitutes a good contribution to the field.

The title is relevant and informative and the abstract matches the whole text. It indicates the main topics, results, and the research question is for me clearly outlined. The introduction is concise and good but needs some updated references.

Regarding the Material and Methods Section, Why the authors placed it after the Results and discussion sections?

For the illustrations, please add titles to the y-axis and x-axis in figure 1 and the x-axis in figure 3.

The design and statistical analysis described in the methods section are in accordance with the objective of the study.

 Line 202, italicize the “L. infantum

Ethics: The protocol of this study was reviewed and approved by Ethics Committee.

The Conclusion section is missing.
